# Generalized local fractions – a method for the calculation of sensitivities to emissions from multiple sources for chemically active species, illustrated using the EMEP MSC-W model (rv5.5)

Peter Wind[1,2] and Willem van Caspel[1]

[1]Climate Modelling and Air Pollution Division, Research and Development Department, Norwegian Meteorological Institute (MET Norway), PO 43 Blindern, 0313 Oslo, Norway
[2]Department of Chemistry, UiT - The Arctic University of Norway, N-9037 Tromsø, Norway

**Correspondence:** Peter Wind (peter.wind@met.no)

**Abstract.**

This paper presents an extension of the original Local Fraction methodology, to allow the tracking of the sensitivity of chemically active air pollutants to emission sources. The generalized Local Fractions are defined as the linear sensitivities of chemical species to source emission changes, as propagated through the full set of non-linear chemical transformations. The method allows to track simultaneously sensitivities from hundreds of sources (typically countries or emission sectors) in a single simulation. The current work describes how the non-linear chemical transformations are taken into account in a rigorous manner, while validating the implementation of the method in the European Monitoring and Evaluation Programme (EMEP) Meteorological Synthesizing Centre – West (MSC-W) chemistry-transport model by examples. While effectively producing the same results as a direct 'brute force' method, where the impact of emission reductions of each source has to be computed in a separate scenario simulation, the generalized Local Fractions are an order of magnitude more computationally efficient when large numbers of scenarios are considered.

## 1 Introduction

Air pollution negatively impacts both human and ecosystem health (Manisalidis et al., 2020), while being the product of a complex interplay between chemistry, meteorology, and natural and anthropogenic emissions. One of the fundamental questions in air quality research is to relate the different emission sources with air pollution in certain receptor regions (see for example The Forum for Air Quality Modelling (FAIRMODE, Thunis et al. (2022)). To this end, the Local Fractions (LF) method was originally developed as a practical way to track the relative contributions to primary air pollution from local and regional sources (Wind et al., 2020). Within the LF framework, tracking a set of sources simultaneously greatly increased the numerical efficiency, allowing the tracking of thousands of (distant) source contributions within a single simulation. However, the method did not consider the effects of chemistry, making it suitable only for chemically inert species. In this work, the LF methodology is generalized to include chemistry, allowing also the description of source sensitivities for species that undergo complex chemical processes.

The original LF method worked by assuming linearity, in the sense that the fractions computed for each source or pollutant can be considered independently. That is, the effect of an increase of emission from two sources being equal to the sum of the effects obtained by increasing each source individually. However, many important pollutants have a non-linear dependency on emissions, while also being the product of chemical reactions rather than being emitted directly. The species involved will also usually originate from different emission sources. To track the source impacts to species such as ozone ($O_3$) and Secondary Inorganic Aerosol (SIA), the LF method is generalized to instead track the *sensitivity* (i.e., rate of change) of such active chemical species to source emission changes. The approach of tracking these chemical sensitivities allows for a mathematically well-defined description of non-linear processes, while also allowing for the quantification of the impact of emission changes of those sources to the total air pollutant. In this work the word "source" is defined as a set of emitted species from a predefined region and from any individual or combination of emission sector(s).

In effect, the sensitivities obtained through the generalized LF method define the changes in pollutant concentrations that would result from a small change in emissions. By 'small', we mean that the method calculates pollutant perturbations arising from infinitesimally small emission perturbations, thereby tracking the tangent of concentrations expressed as a function of emission intensity. Since the emission of a species will affect an entire group of species through non-linear chemical reactions, the sensitivities of all species that are directly or indirectly involved in the chemical processes must be tracked separately for each source. However, despite this added complexity, the method still allows for the tracking of hundreds of sources in a single simulation, building on the efficient formulation of the original LF methodology.

One such applications arises in the calculations of the country-to-country Source-Receptor (SR) contributions to air pollution, which is a standard product of the European Monitoring and Evaluation Programme (EMEP) Chemistry-Transport Model (CTM) developed at the Meteorological Synthesising Centre - West (MSC-W). The CTM developed at MSC-W (hereafter "EMEP model") is a three-dimensional Eulerian model for tropospheric chemistry, having a long history of policy application and research development on both the European and global scales (e.g., Ge et al., 2024; van Caspel et al., 2024; Jonson et al., 2018; Simpson et al., 2012). Traditionally, the country-to-country SR contributions, or "blame-matrices", have been calculated using a large number of Brute-Force (BF) simulations (around 50 per pollutant species), where country emissions for primary particles, nitrogen oxides ($NO_x = NO + NO_2$), Non-Methane Volatile Organic Compounds (NMVOC), sulfur oxides ($SO_x = SO_2 + SO_4$) or ammonia ($NH_3$) are individually reduced by 15 %. The information about the relative impact from different countries (and possibly emission sectors) can then be used to propose an optimized set of abatement measures, as is done for example using the GAINS (Greenhouse gas – Air pollution Interactions and Synergies) model (Amann et al., 2011). With the generalized LF methodology, the impact of emission reductions for all individual countries and precursor species can be calculated from a single simulation.

The methodology is a sensitivity method. It does not directly attempt to determine the impact of any finite change of emissions. Neither does it assign the relative total contributions from different sources (in contrast to other tagging methods (e.g., Butler et al., 2018; Emmons et al., 2012; Dunker et al., 2002; Kwok et al., 2015; Grewe, 2013; Grewe et al., 2017; Wang et al., 1998; Wu et al., 2011)). Total contributions can only be directly inferred if the species are considered to have a linear dependency on emissions, which is not the case in general. Still, due to the lower computational cost, non-linear responses

can be inferred by performing sensitivity analysis at several emission levels, thus providing indirectly the effect of non linear changes (EMEP Status Report 1/2024, 2024, chapter 6,).

The generalized LF methodology (hereafter interchangeably referred to as "LF") is introduced and described in more detail in Sect. 2. The latter includes an illustrative example of the core ideas behind the method through its implementation for the effects of dry deposition and SIA, followed by a more general mathematical description. Sect. 3 discusses the computational cost. Sect. 4 validates the methodology through comparison against the BF method for several model configurations, in terms of numerical accuracy. The results are concluded in Sect. 5, where also a number of future applications for the generalized LFs

are discussed.

## 2    Generalized Local Fractions methodology

In a scenario (or BF) approach, an independent simulation is performed for each source of interest. Conceptually, the core of the Local Fractions method is to perform a single simulation for a set of scenarios (up to a few hundred), and perform the computations in such a way that some key variables can be reused for all scenarios instead of being computed independently

for each simulation. The results from the generalized Local Fractions method are in principle the same as the results from a series of BF scenario runs with small perturbations in the emissions, but can be obtained at a lower computational total cost, if the number of scenarios is larger than about ten (Sect. 3.3.2).

For example, some variables are independent of emission intensity. Meteorological data is an obvious example, since these fields are independent of the emission scenarios, but are nevertheless read in and processed for each individual simulation. In

the Local Fractions method these fields have to be computed only once to cover all the scenarios.

However, the largest benefits arise from avoiding the repetition of computationally expensive processes such as advection and chemical transformations. In the case of advection, the flux for a pollutant is computed only once in the LF method, and can then be reused for all the tracked scenarios. If the different scenarios only differ by a small perturbation (for example, 1 % emission intensity), second order effects are negligible, and this can be taken advantage of to efficiently calculate the effect

of complex transformations. To this end, we will show that computing the Jacobian of the chemical transformations allows for a more efficient treatment of the sensitivities to emission perturbations, avoiding the calculation of the explicit chemical transformations for each individual scenario.

The fundamental theory underlying the generalized Local Fractions is not new within the context of Numerical Weather Prediction (NWP) and chemistry-transport modelling. It is called the tangent linear model, already conceptually introduced by

Lorenz (1965), relating output variables with small perturbations of the initial state. To our knowledge, the usage of Tangent Linear Models (TLM) has primarily been to compute backward trajectories (emission inversions (Zheng et al., 2024)), while the adjoint of TLMs is also applied in the context of data assimilation (Shankar Rao, 2007). We will here show how a TLM can be used in a 'forward mode' to compute sensitivities to emission sources in the form of, for example, source-receptor matrices. We will give a description from an applied point of view in the context of CTMs, utilizing the efficient formulation of the

original Local Fractions.

## 2.1 The original Local Fractions

As described in Wind et al. (2020), the original LF methodology was designed to track the fraction of the total pollutant with an origin from a specific source $k$. The source $k$ can refer to any specific class of pollutants (from a certain sector, or in principle any other sub-class of a pollutant) either from a given grid cell within the tracked region (termed the 'local region') or from a predefined fixed region, such as a country. In the following, we will use upper cases (Local Fractions or LFs) when referring to the methodology used in computing the local fractions, and lower cases (local fractions) for the physical quantity that gives the actual fraction of a pollutant (though this definition changes when considering the generalized Local Fractions, as will be discussed in the following section).

The original LFs defined the local fractions as the dimensionless quantity

$$\text{lf}_k = \frac{C_{i,k}}{C_i} = c_{i,k}, \tag{1}$$

where $C_{i,k}$ represents the concentration of pollutant $C_i$ from source $k$ and we use the lower case $c_{i,k}$ to define the "dimensionless concentration", or the proportion of the pollutant that originate from source $k$. The source $k$ in this notation, refers to a source of pollutant $C_i$ (units of µg m$^{-3}$ for example). For instance $i$ could refer to primary particulate matter, and $k$ could refer to primary particulate matter from the transport sector in Paris. Since the original LF method did not consider the effects of chemistry, Eq. 1 applies only to chemically inert species such as primary Particulate Matter (PM). For such species, considering the emissions ($E$) of pollutant $C_i$ by source $k$ (written as $E_k$, unit µg m$^{-3}$s$^{-1}$), the local fractions at time $t + \Delta t$ can be written as

$$\text{lf}_k(t + \Delta t) = \frac{C_{i,k}(t) + E_k(t)\Delta t}{C_i(t + \Delta t)}, \tag{2}$$

where $C_i(t + \Delta t)$ is the total pollutant concentration calculated by the CTM based on all emission sources. In effect, Eq. 2 then also represents the fraction by which the total pollutant concentration would be reduced if source $E_k$ is omitted (assuming linearity). One practical advantage of using the formulation of Eq. 2 is that physical processes such as wet and dry deposition affect the total pollutant concentrations, but not the fractional source contributions. Hence such processes do not have to be calculated for each individual scenario, or source, but only once.

## 2.2 Sensitivity to emission changes

The conceptual idea behind the generalized Local Fractions is introduced here. We can rewrite $E_k$ using a scalar multiplicative factor $e_k$, which can be used to define a uniform reduction factor on $E_k$, which itself is fully space and time dependent. This can be written as

$$\tilde{E}_k(x, y, z, t, e_k) = e_k E_k(x, y, z, t), \tag{3}$$

where the "base case" with full emissions corresponds to $e_k = 1$. In calculating the sensitivities to emission changes, the $e_k$ values are perturbed to calculate the resulting impacts on the full set of chemical species included in the CTM. The index $k$ thus

describes a particular scenario where the emissions from source $k$ are modified. Written in mathematical form, the definition of the generalized Local Fractions is then defined as the sensitivity $S$, as

$$S(C_i, e_k E_k) = \frac{\partial C_i}{\partial e_k} = \lim_{\epsilon \to 0} \frac{C_i(e_k E_k) - C_i((e_k - \epsilon) E_k)}{\epsilon}. \tag{4}$$

In a BF approach, $\epsilon$ is a fixed fraction (typically 15%), and both terms in the numerator are computed using two independent simulations. In the generalized LF framework, the derivative in the case of a linear system instead becomes

$$\lim_{\epsilon \to 0} \frac{C_i(e_k E_k) - C_i((e_k - \epsilon) E_k)}{\epsilon} = \lim_{\epsilon \to 0} \frac{C_i(e_k E_k) - C_i(e_k E_k) + \epsilon C_i(E_k)}{\epsilon} = C_i(E_k), \tag{5}$$

where $C_i(E_k)$ represents the concentration of pollutant $C_i$ resulting from emissions from source $k$. For linear processes, the definition of Eq. 4 is thus equivalent to the original LF definition of Eq. 1, multiplied by the total pollutant concentration:

$$\frac{\partial C_i}{\partial e_k} = c_{i,k} C_i(t). \tag{6}$$

This shows that the normalization with respect to $\epsilon$ is such that the sensitivities of Eq. 1 give the concentration change extrapolated linearly to a 100% change of emissions. However, the original interpretation as the fraction of pollutant originating from a specific source cannot be used anymore if non-linear (chemical) transformations are involved, even though the calculated sensitivities themselves reflect linearly extrapolated perturbation impacts (i.e., calculated from the tangent from an otherwise complex set of non-linear transformations). The equations and code valid for the original LFs can still be used, however, as will be discussed in the following.

It is important to keep in mind that even if the results are presented in units of concentrations for a 100% change of emissions, they can not be interpreted as total contributions for non-linear species. The values must be interpreted as sensitivities to small emission changes. Those sensitivities can be both positive or negative, and will in general not sum up to total concentrations.

### 2.3 Illustrative example: Dry deposition

Before describing the mathematical formulation of the general case in more detail, we will show in a simpler case how the LF method fundamentally differs from a mere parallel run of scenarios. To that end, we will describe the procedure for computing the contributions of different sources to dry deposition.

The deposition process of species $C_i$ can be written as the product of the concentration of species $C_i$ and an effective deposition velocity $v_i$, as

$$\text{Dep}_i = v_i C_i. \tag{7}$$

Here the effective deposition velocity is taken as a dimensionless number between 0 and 1 (with $v_i = 1$ meaning that all of species $C_i$ is lost to deposition):

$$C_i(t + \Delta t) = C_i(t) - v_i C_i(t). \tag{8}$$

With this notation, $\text{Dep}_i$ has units of $\mu\text{g m}^{-3}$ per timestep, and must be multiplied by the grid cell volume to get the weight of pollutants deposited during the time step. The effective deposition velocity will depend on many parameters, such as fractions of

land use, leaf area index, meteorological parameters, etc. The computation of $v_i$ can therefore be computationally demanding. In a parallel scenario approach, each scenario would recompute the value of $v_i$, and the computation time for those terms is therefore proportional to the number of scenarios. If we instead look at the sensitivity of $\text{Dep}_i$ for species $C_i$ to emission changes from source $k$:

$$\frac{\partial \text{Dep}_i}{\partial e_k} = \frac{\partial v_i C_i}{\partial e_k} = v_i \frac{\partial C_i}{\partial e_k}, \tag{9}$$

the $\frac{\partial C_i}{\partial e_k}$ term is the sensitivity calculated using the generalized Local Fractions, where we have assumed that the deposition velocity $v_i$ is independent of the concentrations for small concentration perturbations. This shows that the deposition velocity has to be computed only once, and the sensitivity to emission changes of the deposition, can be computed with a simple multiplication for each scenario $S_k(C_i)$. The latter being a shorthand notation of the sensitivity of species $C_i$ to source $k$ given by the definition of Eq. 4.

Another aspect of the original LFs was that dry deposition does not change the values of the local fractions itself. That is, deposition changes the total modeled concentrations, but not the fractional contributions of different emission sources tracked by the LFs. Using the dimensionless notation (see Eq. 1), for the deposition process:

$$c_{i,k}(t + \Delta t) = c_{i,k}(t). \tag{10}$$

This property is maintained in the generalized LFs framework, since the generalized LFs are equivalent to the original LFs but multiplied by the total pollutant concentration, Eq. 6. Therefore, while the process of dry deposition affects the total pollutant concentration, the value of the sensitivities (or generalized local fractions) remains unchanged except for the renormalization:

$$\frac{\partial C_i}{\partial e_k}(t + \Delta t) = \frac{\partial C_i}{\partial e_k}(t) \frac{C_i(t + \Delta t)}{C_i(t)}. \tag{11}$$

While Eq. 11 reflects dry deposition during time step $\Delta t$, the handling of other processes during $\Delta t$ (such as chemical transformations) will be discussed later on.

### 2.3.1 Non-linear deposition

So far the results are simply proportional to the magnitude of the linear emission changes. It has been shown Fowler et al. (2001) that for example $SO_2$ deposition is affected by $NH_3$ concentration. If we want to describe such situations where the deposition velocity depends on the concentration of other species $C_j$, where $j$ can represent any number of species (also $j = i$), we need to add this dependency as an additional transformation in Eq. 9, as

$$\frac{\partial \text{Dep}_i}{\partial e_k} = \frac{\partial v_i C_i}{\partial e_k} = \sum_j \frac{\partial v_i}{\partial C_j} \frac{\partial C_j}{\partial e_k} + v_i \frac{\partial C_i}{\partial e_k}. \tag{12}$$

However, the range of validity is now limited, because Eq. 12 assumes that $\frac{\partial v_i}{\partial C_j}$ is independent of the scenario, which is valid only in a first order approximation (i.e., that the scenarios only differ slightly). For larger deviations from the base concentrations, this may not be the case anymore, and the calculated deposition sensitivities may no longer be representative.

Compared to a regular simulation, the additional computational cost will now include the calculation of the derivatives $\frac{\partial v_i}{\partial C_j}$. Note however that this additional cost is still independent of the number of scenarios (indexed by $k$) considered in the LF simulation, such that including additional scenarios comes at practically no additional computational cost.

## 2.4 Chemistry

In Wind et al. (2020), we showed how the local fractions are transformed during advection and other linear processes, largely
analogous to the dry deposition example from the previous section. In this section we will show how they are transformed through non-linear chemical transformations. But before presenting the general equations for all species and relevant model process sensitivities, we will show in more details how the chemical transformations of SIA are treated within the generalized LF framework. This will illustrate in a simpler context the way the generalized LFs are transformed when chemical transformation are taking place.

### 2.4.1 Generalized Local Fractions approach for SIA

In the thermodynamic equilibrium chemistry modules of CTMs, the concentrations of the species $HNO_3$, $NO_3$, $NH_4$, $NH_3$ and $SO_4$ are partitioned into the gas and particulate phases, with a dependence on each other as well as on physical environment variables such as temperature and humidity.

We consider one grid cell, and the chemical transformation during one time step. In a direct scenario approach, a set of
195 concentrations is considered for each scenario. For each scenario, the following transformation is applied:

$$C_i^k(t + \Delta t) = f_i(\{C_j^k(t)\}),\tag{13}$$

where $C_i^k$ are the concentrations, $k$ is the scenario index and $i$ and $j$ run over the five species $HNO_3$, $NO_3$, $NH_4$, $NH_3$ and $SO_4$. $f_i$ represents the non-linear thermodynamic equilibrium transformation. In this formulation the equilibrium module is applied for each scenario, and the computational cost will then be proportional to the number of scenarios.
We now assume that all the scenarios differ by only small amounts compared to a "base case" $C_i$:

$$C_i^k(t) = C_i(t) + \delta C_i^k(t).\tag{14}$$

In a linearized picture, where we assume that $\delta C_i^k$ are small, the resulting concentrations at time step $t + \Delta t$ can be approximated to first-order using a Taylor-series expansion as

$$C_i^k(t + \Delta t) = C_i(t + \Delta t) + \sum_j^5 a_i^j \delta C_j^k(t),\tag{15}$$

where $C_i$ is the concentration of one of the five species ($i = 1, ..., 5$) and $a_i^j$ are the transformation parameters, which describe the transformation of the concentrations of species $C_i$ as a function of species $C_j$ during time step $\Delta t$. $a_i^j$ will depend on the environment, the size of the time step, but also on the base case concentrations; however in a linearized picture, $a_i^j$ are considered constants during the time step, being independent of the scenario $k$.

The clue here is that the matrix $a_i^j$ is of a fixed size $(5 \times 5)$, independent of the number of scenarios $k$. The number of operations required in Eq. 15 is still proportional to the number of scenarios. However, the number of operations for each scenario is very small (five multiplications, and six additions). By comparison, the application of the full equilibrium module typically requires thousands of operations.

To compute the updated values of the generalized local fractions at time $t + \Delta t$, the $\delta C_j^k(t)$ terms in Eq. 15 can be chosen to be proportional to the sensitivities:

$$\delta C_j^k(t) = \delta \frac{\partial C_j}{\partial e_k}. \tag{16}$$

Then Eq. 15 can be written as

$$C_i^k(t + \Delta t) = C_i(t + \Delta t) + \delta \frac{\partial C_i}{\partial e_k}(t + \Delta t) = C_i(t + \Delta t) + \sum_j^5 a_i^j \delta \frac{\partial C_j}{\partial e_k}(t), \tag{17}$$

and thus

$$\frac{\partial C_i}{\partial e_k}(t + \Delta t) = \sum_j^5 a_i^j \frac{\partial C_j}{\partial e_k}(t). \tag{18}$$

This shows that each updated local fraction can be computed directly using only the matrix $a_i^j$, without having to explicitly apply the full transformation of Eq. 13 for each source $k$.

We still need to compute the matrix $a_i^j$. This can be done in practice by evaluating Eq. 13 for five additional sets of input values, where in each set only one of the pollutants is perturbed:

$$C_j^j(t) = C_j(t) + \delta C_j(t) \tag{19}$$

$$C_i^j(t) = C_i(t) \qquad i \neq j. \tag{20}$$

We get then one column of the matrix for each iteration by inverting the equation:

$$a_i^j = \frac{C_i^j(t + \Delta t) - C_i(t + \Delta t)}{\delta C_j(t)}, \tag{21}$$

where $C_i^j(t + \Delta t)$ is the result for species $i = 1, ..., 5$ from the calculation perturbing species $C_j$ and $C_i(t + \Delta t)$ is the result from the unperturbed (base) case.

Alternatively one can use dimensionless local fractions $\frac{\partial c_j}{\partial e_k} = \frac{\partial C_j}{\partial e_k} \frac{1}{C_j}$, as is done in our code implementation for the EMEP model:

$$b_i^j = a_i^j \frac{C_j(t)}{C_i(t + \Delta t)} = \frac{C_i^j(t + \Delta t) - C_i(t + \Delta t)}{\delta C_i(t + \Delta t)}. \tag{22}$$

The updated values can then computed in the same way:

$$\frac{\partial c_i}{\partial e_k}(t + \Delta t) = \sum_j^5 b_i^j \frac{\partial c_j}{\partial e_k}(t). \tag{23}$$

The reason for using dimensionless local fractions in the code itself stems from historical reasons with simulations of only primary PM.

We note that in the EMEP model, the resulting equilibrium solutions as calculated with the MARS thermodynamic equilibrium module (as used in the current work) depends only on the sums of $HNO_3 + NO_3$ and $NH_4 + NH_3$, and on the $SO_4$ concentrations at time $t$. Therefore only three supplementary evaluations of Eq. 13 are actually required to determine the entire
$a_i^j$ matrix.

In these calculations $\delta$ is chosen to be a fixed small number, normally $\delta = 0.05$. Mathematically a very small value can be chosen, however for numerical stability and robustness reasons in the SIA calculations, a somewhat larger value than a 1 % perturbation is preferred here.

### 2.4.2 General case

Given the generalized local fractions in a given grid cell at time $t$, we want to compute the new values after that the concentrations have been updated through the Chemical module of the host CTM. We will assume (as is the case in our model code), that the emissions are included as additional terms in the Chemistry module.

The concentrations at time $t + \Delta t$ can be expressed as a general function of all the input concentrations and emissions at time $t$. We will only write explicitly the parameters that are affected by the emission sources (e.g., $\Delta t$, temperature, etc., are not
modified by a change in emissions in our model, and are therefore not included). If some parameters are emission dependent (for example, solar radiation may depend on aerosol concentrations, and hence on emissions), they can be included in a similar way as discussed in the following.

Writing the concentrations at time $t + \Delta t$ gives

$$C_i(t + \Delta t) = f_i(C_1, C_2, ..., C_{n_C}, E_1, E_2, E_3, ..., E_{n_E})(t), \tag{24}$$

where all concentrations on the right-hand side are at time $t$ and $E_j$ are the emissions sources. If we derive the equation with respect to $e_k$ (assuming that $f_i$ and its partial derivatives are continuous functions) we get:

$$\frac{\partial C_i}{\partial e_k}(t + \Delta t) = \sum_j^{n_C} \frac{\partial f_i}{\partial C_j} \frac{\partial C_j}{\partial e_k}(t) + \sum_j^{n_E} \frac{\partial f_i}{\partial E_j} \frac{\partial E_j}{\partial e_k}, \tag{25}$$

where $\frac{\partial C_j}{\partial e_k}(t)$ and $\frac{\partial C_j}{\partial e_k}(t + \Delta t)$ are the values of the sensitivities at time $t$ and $t + \Delta t$. In Eq. 25, the $\frac{\partial f_i}{\partial E_j}$ and $\frac{\partial f_i}{\partial C_j}$ terms define the Jacobian of the transformation, $\frac{\partial f_i}{\partial E_j}$ represents the average rate of change in the concentration $C_i$ due to the emissions $E_j$
during $\Delta t$. $\frac{\partial E_j}{\partial e_k}$ shows the dependence of the emissions of species $j$ to source $k$; typically this could be 1 within a country referenced by $k$ that emits species $j$, and zero outside of the country; it could also be some other fraction if one looks at more complex situations such as sector specific emissions.

Note that $\epsilon$ in Eq. 4 is assumed small, however $\Delta t$ or the changes in Eq. 24 are not assumed to be small. In the general case, and in the model code, the function $f$ is not assumed to be linear, and Eq. 25 is valid also for non-infinitesimal $\Delta t$. For
example, in the chemical solver $\Delta t$ is divided into micro-iterations, each time step capturing the full non-linear chemistry.

If in a chemical scheme the chemical transformations can be assumed linear within the time step, the Jacobian might be directly accessible as an analytical function of the input concentrations. Otherwise, the computation of the Jacobian matrix must be computed for example by a method similar as shown for SIA in Sect. 2.4.1.

In our present implementation, 60 species will have a direct or indirect effect on other chemical species in the chemistry module. This implies that the function $f$ must be evaluated for a perturbation of each of those 60 species. In addition perturbations for emissions must be performed, but usually only a few sources will contribute for a given grid cell (typically one country, four species, and only for the lowest seven vertical levels). The evaluation of the function $f$ for each of these perturbations represents the most time-consuming part of the entire code (see Sect. 3.3).

Eq. 25 is a general equation that describes how to update the sensitivities for any transformation. For example in the (linear) deposition case, the Jacobian is simply equal to $1 - v_i$. In the advection case (below) the indices of the concentrations would refer to pollutants at different position in space and the Jacobian will be equal to the fluxes.

## 2.5 Advection

Mathematically the advection can be treated similarly in the Local Fraction method. Now the sensitivity updates can depend on the values at different positions in space, but there is no mixing between different species caused by advection.

In a one-dimensional advection time step, some pollutants are transferred from one cell to a neighboring cell, which can be written as

$$C_i(x_0, t + \Delta t) = C_i(x_0, t) + F_i(x_{-\frac{1}{2}})C_i(x_{-1}, t) - F_i(x_{+\frac{1}{2}})C_i(x_{+1}, t). \tag{26}$$

Here $F_i(x_{-\frac{1}{2}})$ and $F_i(x_{+\frac{1}{2}})$ are unitless fluxes through the cell boundaries; they represent the fraction of pollutants transferred between two neighboring cells. These fluxes are the fraction of air masses transported to a neighboring grid cell and are in reality independent of the pollutant concentrations. In a simplified zero order scheme the fluxes would simply be proportional to the wind speed $\frac{v\Delta t}{\Delta x}$, with $v$ wind speed and $\Delta x$ the size of the cell. However, in the EMEP model the one-dimensional Bott's fourth order scheme (Bott, 1989a, b) is used. The main purpose of the scheme is to minimize the so-called numerical diffusion. In effect, using the Bott scheme means that the fraction of pollutant that is transferred between neighboring grid cells will depend not only on the wind, but also on concentrations of five surrounding grid cells and thus indirectly on emissions. This has consequences when a brute force method is applied: the changes in concentrations observed when emissions are modified, are not only a consequence of physical effects. The fluxes calculated in two scenarios will be different and this will affect the transport patterns, because the advection scheme will be based on different concentration distributions and this will have an indirect, non-physical effect. One visible adverse consequence of this effect is that in a BF approach, it can sometimes be observed that an increase of emissions in a grid cell, can lead to a decrease of concentrations in an upwind grid cell.

Using the formalism from the preceding section (Eq. 25), Eq. 26 becomes:

$$\frac{\partial C_i}{\partial e_k}(x_0, t + \Delta t) = \frac{\partial C_i}{\partial e_k}(x_0, t) +$$

$$\sum_n \frac{\partial F_i(x_{-\frac{1}{2}})}{\partial C_i(x_n)} \frac{\partial C_i(x_n)}{\partial e_k}(t) C_i(x_{-1}, t) + F_i(x_{-\frac{1}{2}}) \frac{\partial C_i(x_{-1})}{\partial e_k}(t) +$$

$$\sum_n \frac{\partial F_i(x_{+\frac{1}{2}})}{\partial C_i(x_n)} \frac{\partial C_i(x_n)}{\partial e_k}(t) C_i(x_{+1}, t) + F_i(x_{+\frac{1}{2}}) \frac{\partial C_i(x_{+1})}{\partial e_k}(t), \tag{27}$$

where $n$ runs over the five neighboring cells and $\frac{\partial C_i(x_n)}{\partial e_k}$ is the sensitivity of species $C_i$ to source $k$ in cell $x_n$. The $\frac{\partial F_i}{\partial C_i}$ terms reflect the dependence of the fluxes on the concentrations of neighboring cells. If we would use a simpler scheme where the fluxes do depend only on wind speed, and not on the concentrations, $\frac{\partial F_i}{\partial C_i} = 0$ and the corresponding terms in Eq. 27 do not contribute (this is the case for the zero order advection approximation used for comparisons in Sect. 4).

In principle we could use Eq. 27 in the generalized Local Fractions framework, and reproduce also the results arising from advection changes from BF in the case of small emissions changes. Instead, for the horizontal advection of the sensitivities, or local fractions, we will set $\frac{\partial F_i}{\partial C_i} = 0$. That is, use the same base fluxes $F_i$ for all the scenarios $k$, and not take into account the changes in the fluxes that appear when concentrations change. Eq. 27 can then be written as

$$\frac{\partial C_i}{\partial e_k}(x_0, t + \Delta t) = \frac{\partial C_i}{\partial e_k}(x_0, t) + F_i(x_{-\frac{1}{2}}) \frac{\partial C_i(x_{-1})}{\partial e_k}(t) + F_i(x_{+\frac{1}{2}}) \frac{\partial C_i(x_{+1})}{\partial e_k}(t), \tag{28}$$

which gives a more stable transport pattern and ensures that for primary species an increase of emissions can never give a decrease in concentrations in the advection process. The $F_i$ fluxes in Eq. 28 are then simply the same as those calculated by the CTM for the regular advection of species.

For the vertical advection, the EMEP model uses a second order scheme. In that case we have still chosen to include all terms of Eq. 27. This is because $O_3$ has high values at high altitude, and this can have a strong effect on the vertical transport patterns . To keep a better compatibility with the BF method, we have found it preferable here to not use only the base fluxes, as in the horizontal advection case.

## 3   Computational aspects

The LF modules are additions to the original EMEP model code, but do not affect the original results (e.g., those used for BF calculations). For each module (advection, chemistry, emissions, depositions etc.) a corresponding LF module exists, which at each time step computes the updates to the local fractions, but there is no feedback from the LF modules into the regular concentrations.

### 3.1   Differences with BF approach

In theory, the generalized Local Fractions method can give results identical to a direct method when two runs which differ only by a small change of emissions are compared. In practice some differences are still present. The largest differences are due to

the differences in the treatment of advection. As discussed in the preceding section, this is expected to slightly *improve* the quality of the results rather than deteriorate them. For testing purposes, it is still possible to use a simplified advection scheme ("zero order" Bott scheme), for which the results from the advection module using both methods will be identical.

Some transformations (i.e., partial derivatives) are not yet implemented. For example some reaction rates depend on the surface of particulate matter present in a grid cell. A change of emissions may affect the size of the particulate matter, and thereby give a change of those reaction rates. These secondary effects are at present not taken into account in the local fraction calculations.

Another limitation stems from the photolysis rate ($J$-value) calculations made using the Cloud-$J$ module, which is the
default photolysis rate scheme used by the EMEP model from version 4.47 onward (van Caspel et al., 2023). Since Cloud-$J$ takes into account the instantaneous modeled abundance of $O_3$ and a number of aerosol species throughout the atmospheric column, the chemistry inside a single grid cell is no longer completely local, depending also on the radiative impact of chemical concentrations (and perturbations) in the above grid cells. The impact of this limitation is however expected to be comparatively small, with the majority of over-head absorption of radiation relevant to active chemistry occurring above the EMEP model top
(100 hPa), for which (UV-absorbing) $O_3$ concentrations are specified based on observations.

There are other differences due to the details of the numerical schemes. One example, is the scheme for chemical transformations: the chemical scheme uses a fixed number of iterations. The starting guess will depend on the concentrations from the previous time step. For the local fractions, the starting guess is also taken from the corresponding scenario at the previous iteration, however this is not completely equivalent.

## 3.2  Filtering

Some processes may present discontinuities, where an infinitesimal change in an input value can produce a non-infinitesimal change in the output values. This it not uncommon in for example two situations:

- A test is be performed on the value of a concentration (chemical regime), and the code can branch into one transformation in one case and into another for the other case.

- An iterative procedure is be used, and the number of iterations used depends on some concentration-dependent criterion. The number of iterations may then vary slightly between otherwise almost identical cases.

In a LF approach, if numerical derivation is used, the chance of being just at the two sides of a branching point is small, but the effect will also be large, since the derivative value is obtained by dividing by $\epsilon$ in Eq. 4. In a BF approach these discontinuities may happen more often, but their effect is also smaller. Such discontinuities have been observed in thermodynamic equilibrium
chemistry modules (e.g., Capps et al., 2012), as will also be discussed in Sect. 4.3.

One advantage of the LF approach is that it is possible to filter out such effects, if they are not too numerous: since in those cases the calculated derivatives will be much larger than can reasonably be expected, they can be detected and some action taken (simply keep the values of the local fractions unchanged for this particular point and time step for example). Filtering the results is much more difficult in a BF approach, since it is difficult to recognize the problematic situations: since the BF base

run and scenario run are independent, it is not possible to detect those special situations. It is also not trivial to define what to do if one would detect a problematic chemical regime.

### 3.3 Computational cost

Since the main advantage of the LF method is its computational cost, we will present in some details how the cost compares to direct scenario runs.

#### 3.3.1 Cost of chemistry

The computation of the chemical transformations is the computationally most expensive part of the EMEP model; this is probably the case for most CTMs. The calculation of the Jacobian matrix will increase this cost substantially and is the computationally most demanding part of the LF calculation. This cost is however in theory independent of the number of pollutant sources that are traced. This is a fundamental difference compared to direct methods. For those methods, the number

of components that undergo the full chemistry scheme increase proportionally with the number of scenarios.

The number of operations performed in Eq. 25 will still increase with the number of traced pollutants, but it has the form of a matrix multiplication. Matrix multiplications can be done extremely efficiently on most computers, and will in practice have a negligible computational cost.

The computation of the Jacobian matrix is a fully local process (i.e., local to each grid cell), and with our code it will scale

perfectly with the number of Message Passing Interface (MPI) processes in a multi-core parallel run. This means that the computation time can in practice be reduced by increasing the number of processors used.

#### 3.3.2 Scaling of computation time with number of scenarios

Table 1 compares the time required to run the EMEP model using the LF method, for different numbers of scenarios. We assume that each source region (country) is analyzed for five different emissions reductions: $SO_x$, $NO_x$, $NH_3$, VOC and primary PM,

and therefore count five scenarios for each country in addition to the baseline scenario. Here we note that some more specific details about the EMEP model itself will be discussed in Sect 4.

When a small number of scenarios are evaluated, the total time for a LF run is almost independent of the number of scenarios. This is because most of the time is spent computing the Jacobian matrix of the chemistry module, and this time does not depend on the number of scenarios. For the cases with large number of scenarios, the computation time is dominated by the

local fractions updates required for each scenario (mainly in the advection module), and the run time increases linearly with the number of scenario. This is reflected by a constant time per scenarios, or speed up.

The speed up is better for around 250 scenarios than it is for higher number of scenarios. That means that it is more efficient to run for instance 250 scenarios twice, than to run 500 scenarios in a single run. This can be explained, because even if the number of operations is formally smaller when doing all the scenarios together, in practice the data arrays become very large; if

**Table 1.** Relative run times of LF simulations and their speed up in comparison to BF. The time unit is defined as the time to run a single BF scenario, amounting to 29.1 seconds ($T_{\text{scenario}}$) for the model setup described in Sect. 4 (48 hours simulation on a $400 \times 260 \times 20$ grid). 50·2, 50·4, and 50·5 stands for 50 countries with 2, 4 or 5 different sector contributions (each emitting five pollutants). The results in the table show the time to execute the LF code on eight compute nodes. The speed up is defined as the time it would take to compute all the scenarios if a direct method (BF) had been used, divided by the time required using the LF method ($\frac{\text{number of scenarios} \cdot T_{\text{scenario}}}{\text{total LF time}}$).

| Number of countries | 2 | 6 | 12 | 25 | 50 | 50·2 | 50·4 | 50·5 |
|---|---|---|---|---|---|---|---|---|
| Number of scenarios | 11 | 31 | 61 | 126 | 251 | 501 | 1001 | 1251 |
| Total LF time / $T_{\text{BF scenario}}$ | 10.5 | 11.0 | 11.0 | 13.5 | 18.3 | 47.9 | 109.0 | 137.1 |
| Speed up | 1.1 | 2.8 | 5.6 | 9.3 | 13.7 | 10.5 | 9.2 | 9.1 |

the number of array elements that are looped over becomes too large, the CPU will run out of memory cache, thereby causing a drop in hardware efficiency. In the future, such limitations should be avoidable by improved code design.

In practice (EMEP Status Report 1/2024, 2024), a full SR matrix calculation for 55 countries on a latitude-longitude-altitude grid with dimensions $400 \times 260 \times 20$ ($0.2° \times 0.3°$ horizontal resolution), will for a full year LF simulation on 16 compute nodes (running 128 MPI processes on each compute node) take 18 wall time hours. This can be compared to a single run on

four compute nodes which requires 2 hour and 40 minutes. A direct BF method would require $(55 \times 5 + 1)$ single runs. The computational resources required would then be approximately 10 times higher compared to the LF run.

Because all species involved in the transformations have to be tracked independently, tracking 55 countries for four chemically active species ($SO_x$, $NO_x$, $NH_3$, VOC) and primary inert particles ($PPM_{2.5}$ and $PPM_{co}$) represent more than 15000 individual local fractions to track.

**3.4   Optimization by approximating chemically active species**

Each term of the Jacobian matrix, $\frac{\partial f_i}{\partial C_j}$ in Eq. 25, is explicitly evaluated in our present implementation. This might not be necessary, and in this section we will briefly indicate the types of simplifications that might be implemented in the future.

The number of Sulfur (S) or Nitrogen (N) atoms are conserved during chemical transformations. We can in principle track the atoms from different sources in a physically meaningful way, and the sum of contributions from different sources will be

equal to the total contribution. The difficulty arises because the atoms are part of different type of molecules. In a given point in time and space, the relative amount of the different molecules ($SO_2/SO_4$, $NO_2/NO$, and $NH_3/NH_4$ for example) will be different for the atoms from different sources, because they have a different history.

In a first approximation one can assume than those relative amounts are source independent. For the Local Fraction method, this would represent a great simplification, as the $SO_x$, $NO_x$, or $NH_x$ molecules can then be treated as primary (as show and

discussed in Wind et al. (2020)). Additional effects can be taken into account if the different molecules are also tracked separately for the different sources. However this will still not be exact, since all chemical species involved in the transformations (such as OH, $O_3$) should all be tracked for completeness.

One could try to further generalize this approach by grouping species into families where the total number of members of a family is conserved during chemical transformations. This can then be combined with a simpler chemical scheme for the computation of the Jacobian matrix.

As explained for SIA, even if the $a$ matrix (Eq. 18) is a $5 \times 5$ matrix, only three additional evaluation of the SIA operator are necessary to compute it, not five as in a general case. For the full chemistry module similar simplifications can be found. In our code the species involved in the ozone chemistry can produce SOA (Secondary Organic Aerosols), but the SOA species do not influence $O_3$, not even indirectly. Meaning that a block of the Jacobian matrix is known to be zero and does not need to be computed. This is taken advantage of in the generalized LF code. The oxidized and reduced nitrogen atoms are conserved during the chemical process, which means that some simplifications could be obtained as for SIA. This is however not implemented yet.

From a more mathematical view point, one can consider the chemical transformations (in one grid cell, during one time step) as a matrix relating small changes in input to change in the output concentrations. This matrix can be diagonalized, and the eigenvectors with eigenvalues with value zero reflect a conserved quantity. The rank of the matrix is then smaller than its size. Building on this, one could keep only the eigenvectors with largest eigenvalues, and neglect the one below a threshold.

There are many more paths to explore that could improve the efficiency of our code, ranging from very simple (updating the Jacobian matrix every second time step only), to purely computational (using a GPU accelerator for the evaluation of the Jacobian matrix).

## 4 Validation

In order to verify that the code actually gives the expected values for the emission sensitivities, the sensitivities calculated using the LF method are compared to the brute force method for 1 % emission perturbations. The latter are sufficiently small that differences between the two methodologies due to non-linear chemistry are avoided, thereby isolating only the methodological differences. Indeed, if the emissions differences are small enough, the calculated derivative obtained by finite differences (BF) should be equal to the sensitivities calculated with the LFs.

The main EMEP model settings are essentially standard, employing standard EMEP reported emissions, except that some natural emissions are omitted (soil $NO_x$, ocean Dimethyl sulfide (DMS), lightning, forest fires, dust, aircrafts) for simplicity. The model is run on a $0.3° \times 0.2°$ horizontal grid, employing 20 vertical levels up to a model top height of 100 hPa. The meteorology is based on 3-hourly data derived from the ECMWF Integrated Forecasting System (IFS) cycle 40r1 model (ECMWF, 2014). While the EMEP model uses its default EmChem19 mechanism (Bergström et al., 2022), employing a simplified set of lumped VOC species (Ge et al., 2024). As noted before, the following setup also employs the MARS equilibrium chemistry module. The results can be reproduced using the code and data provided under the "Code and data availability" section.

The length of the simulation is only 24 hours, in order to have a light weight setup. The purpose here is only to validate the principle of the methodology, not to quantify the differences in all possible situations (climate, species, time scales, emissions, etc.). However, more extensive and realistic comparisons can be found in EMEP Status Report 1/2024 (2024) Ch. 5 and EMEP

Status Report 1/2023 (2023) Ch. 5. Furthermore, while the current work focuses on the differences in results arising from methodological differences, in practice BF calculations are often performed using 15 % rather than 1 % emission reductions. The impact on the differences between the BF and LF methodologies for 15 % emission reductions are also investigated in more detail in EMEP Status Report 1/2023 (2023) Ch. 5 and EMEP Status Report 1/2024 (2024) Ch. 5, noting that the results are qualitatively similar to those discussed in the following.

In particular, in EMEP Status Report 1/2024 (2024) country source-receptor matrices for yearly averages using the BF and LF methods have been compared for several pollutants and indicators, including $O_3$ and $PM_{2.5}$. The differences are overall less than 10 %, and can be considered as small, since they are of the same order of magnitude as methodological differences (advection scheme and filtering) shown in Fig. 1 and Fig. 2 in the following. The differences due to non-linearities introduced by reducing the emissions by 15 % in the BF method, compared to, in principle, infinitesimally small reductions in the LF methodology, represent only a small fraction of methodological differences. We do stress that the 15 % emission reduction employed by the BF method is in principle arbitrary (EMEP Status Report 1/2004, chapter 4, 2004), and that the differences due to non-linearities with the LFs do not represent an actual source of methodological error.

## 4.1 Country example

In this section, the net impact of 1 % perturbations in $NO_x$, $SO_x$, $NH_3$ and VOC emissions are investigated for Germany (DE). Here DE is taken as being a representative country for the comparison of the LF and BF methods, featuring considerable geographic differences in emissions and chemical regimes. The output species included in the comparison are daily mean $O_3$ and SIA, both involving highly active chemical transformations. However, the LF outputs are in general also available for a range of other species and derived $O_3$-indicators, such as Peak Season (April-September) average Maximum Daily 8-hour Average (MDA8) $O_3$, and reactive nitrogen and sulfur deposition. Here we note that, in order to calculate the impact of a 1 % emission reduction using the LF outputs, the LF outputs are multiplied by a factor of 0.01, since the LF outputs represent the impact of emission perturbations linearly extrapolated to a 100 % change of emissions.

As discussed in Sect. 2.5, one source of discrepancies between the BF and generalized LF methods is the (intended) difference arising from the advection scheme used. In order to distinguish differences due to other causes, a set of test runs have also been performed with a simplified advection scheme (zero order advection). In this simplified scheme, there are no differences due to difference in the advections treatment, at the cost of some additional numerical diffusion in both results.

## 4.2 Surface $O_3$

The comparison between the BF and LF methods in the regular setup (fourth order advection) as well as for the zero order advection setup is illustrated in Fig. 1, here shown for the change in surface $O_3$ resulting from the combined perturbations in $NO_x$ and VOC emissions (being those precursor species to which $O_3$ is most sensitive).

In Fig. 1a it can be seen that a reduction in $NO_x$ and VOC emissions leads to decreases in surface $O_3$ almost everywhere inside and in the vicinity of Germany. Nevertheless, due to titration effects parts of Northern Germany and Berlin see an increase in $O_3$ concentrations. Fig. 1b illustrates that these effects are well captured by the LF methodology, while the corresponding

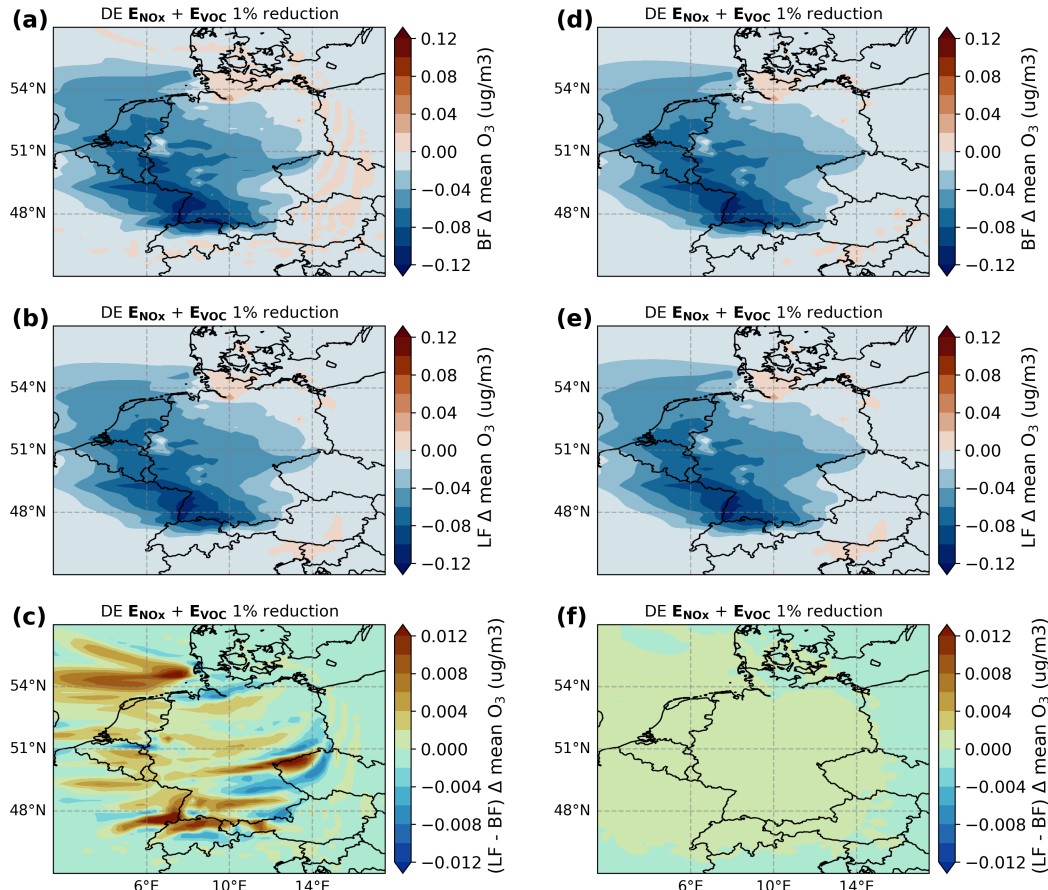

**Figure 1.** Impacts of 1 % $NO_x$ and VOC emission reductions in Germany (DE) on daily mean surface $O_3$ concentrations, calculated for the 1st on July 2018 using the EMEP model with a default fourth order advection (left-hand panels) and zero order advection (right-hand panels) configuration. Top panels show the impacts calculated using the BF method, middle panels with the LF method, and the bottom panels the difference between the two. The scaling of panels (**c**) and (**f**) is a factor of ten smaller than that of the other panels.

zero order advection runs (panels d and e) show very similar results. The difference between the BF and LF results for the fourth
order advection setup (panel c) shows a pattern of positive and negative variations, having a magnitude of around 10 % of that of the individual BF and LF results. However, the lack of difference between BF and LF for the zero order advection scheme (panel f) demonstrates that the differences between the fourth order runs are almost entirely due to the choice of advection scheme. The remaining discrepancies in the zero order advection setup are very small (of the order of 1% in this particular test).

## 4.3 SIA

Fig. 2 shows tests of the impacts on surface SIA concentrations calculated for the combined effects of 1 % reductions in $NO_x$, $SO_x$, and $NH_3$ emissions using the zero order advection setup. While the BF (panel (a)) and LF (panel (d)) calculations show generally agreeable results, their difference (panel (b)) shows a comparatively large difference northwest of the Netherlands. Further diagnostic simulations finds that these differences arise due to gas-aerosol partitioning calculations taking place in the MARS thermodynamics equilibrium module. While the differences between the BF and LF results are still comparatively small, they do point towards the general complication with the use of complex numerical models to calculate the impact of small (emission) perturbations, as discussed in Sect. 3.2.

For example, in the MARS module small perturbations to its input parameters can lead to a change in the number of iterations applied to certain solver routines. Furthermore, certain physical mechanisms, such as aerosol water uptake (in turn affecting the equilibrium solution), can show step-like behavior near certain threshold values. While we have modified the MARS module to smooth the solution in certain parts of its code, variations in the calculated SIA such as those shown in Fig. 2b nevertheless persist. These do, however, not have a large impact on the total simulation results, also for simulations performed over longer time periods.

To further illustrate the effects of numerical instabilities, Fig. 2e shows the difference between BF calculations employing a 1 % emission reduction ($BF^-$) and a 1 % emission increase ($BF^+$). Noting that here the $BF^+$ simulation has been used to likewise calculate the impact of a 1 % emission reduction by changing the sign of its results. One would expect the results between the regular $BF^-$ and $BF^+$ calculations to be almost identical, but in practice they are not, due to numerical effects arising from the complex thermodynamics calculations. This also demonstrates that the problems with discontinuities are present in the BF method too and are not specific to the LF method.

The principle of the SIA filter in the EMEP model is to reject results where more than four molecules in the outputs are created or destroyed for each additional (perturbation) molecule in the input. In Fig. 2c we also show the result for the LF method using an alternative filtering technique. In this filtering method each numerical derivative (Eq. 22) is performed twice, now for one with a positive $\delta$ and also for one with a negative $\delta$. If the ratio between the positive and negative input perturbations does not fall within a factor of three, they are rejected, meaning that the local fractions are kept unchanged in the equilibrium module for this chemistry timestep. Furthermore, when the derivatives calculated with the positive and negative perturbations are found to be in agreement with each other, their geometric mean value is used as the final sensitivity.

Fig. 2f shows specifically the effect of this new filtering technique ($LF^\pm$) by comparing against a LF run without any filtering at all ($LF_{nofilter}$). As expected, the regions that show discontinuities in the BF method (panel (e)) are also regions where the filter is activated. However, there are also regions where the filter has a small effect, without the BF results showing any problem. We nevertheless find that the alternative SIA filter ($LF^\pm$) overall produces the most numerically stable results.

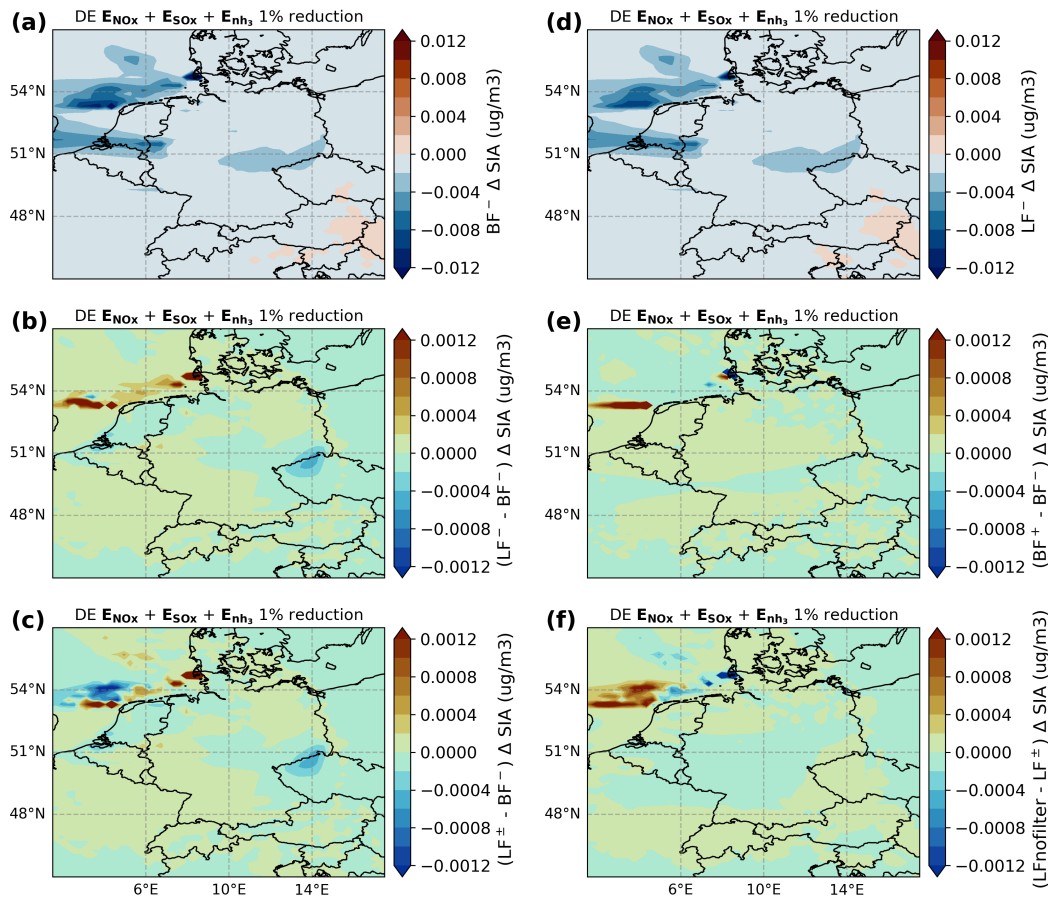

**Figure 2.** Different tests for the impacts of 1 % $NO_x$, $SO_x$ and $NH_3$ emission changes on daily mean surface SIA concentrations calculated for the 1st on July 2018 using the EMEP model with zero order advection. Panel (a): using the brute force method with an emission reduction ($BF^-$). Panel (d): using LF and filtering of number of molecules created in response to a negative input perturbation ($LF^-$). Panel (b): difference between panel (d) ($LF^-$) and panel (a) ($BF^-$). Panel (e): Difference of BF method using a reduction of 1 % ($BF^-$) and an increase of 1 % ($BF^+$). Panel (c) same as (b), but using a different filtering method ($LF^\pm$, see text). Panel (f): Difference using LF with or without filtering.

## 5 Conclusion

The current work describes the theory and implementation of the generalized Local Fractions, which can be used to efficiently track the linear sensitivity to emission changes of air pollutants subject to complex non-linear chemical transformations. Building upon the efficient formulation of the original Local Fractions, the generalized formulation allows the tracking of the sensitivities to hundreds of sources in a single simulation, increasing the computational efficiency by a factor of 10 over the standard SR "blame-matrix" computations performed annually by MSC-W using the EMEP model. While differences between

the emission reduction impacts calculated using the BF and LF methodologies exist, these can largely be understood, arising predominantly as an adverse side effect from the choice of advection scheme in the BF simulations.

The use of the original Local Fractions method has already proven fruitful in several applications considering pollutants as inert particles.

- The uEMEP scheme is a down-scaling scheme (Denby et al., 2020), allowing to describe air pollution at fine resolution (down to 25 m), but still taking into account the effect of long range transport. In this scheme the local fractions gives the fraction of the pollutants which have a local origin, and those can then be replaced by more accurate, fine resolution values (Denby et al., 2024a, b).

- For the GAINS (Greenhouse gas – Air pollution Interactions and Synergies) model (Amann et al., 2011), a full analysis of the SR relationships (from any part to any grid cell) over large regions have been produced using the local fractions (also called "transfer coefficients" in this context). Applications in Europe (Klimont et al., 2022) and South East Asia (World Bank Group, 2023) exist. A report on the methodology used in such applications of GAINS is under preparation.

- Using hourly time tagged emission sources, it is possible to use the LFs for inverse modelling, i.e., trying to reconstruct the emission sources based on observations. Such developments are presently underway.

The generalized Local Fraction calculations with the full chemistry are not efficient enough to give results as detailed as for inert particles (where tens thousand of sources can be tracked simultaneously). Still, when a large number of scenarios are to be simulated, the new approach is much more efficient than previously available methods, opening up new fields of applications which are presently being investigated:

- Due to computational limitations, the country-to-country blame-matrices calculated by MSC-W are normally performed on a reduced resolution $0.3° \times 0.2°$ horizontal grid spanning 30°N-82°N to 30°W-90°E. However, with the considerably more efficient general LF method, future blame-matrix calculations could be performed on the regular $0.1° \times 0.1°$ grid without loss of numerical accuracy.

- The sensitivities to emission source perturbations calculated using the generalized LF method can be used as SR relationship coefficients in the calculation of cost-effective emission control strategies with the GAINS model, also for chemically active species such as $O_3$. Such calculations could furthermore benefit from the use of sensitivities calculated from simulations with different background emission levels, to include a description of the non-linear response to emission reductions when the reductions are comparatively large.

- A complete picture of the SR relationships at different background levels, allows to describe the *accumulated* contributions from each country using the path integral method (Dunker, 2015). By integrating the sensitivities over a given emission change pathway, it is possible to determine the source contributions differences between two emission scenarios (see, e.g., EMEP Status Report 1/2024, 2024, Ch. 6). This allows to relate the source sensitivities with source apportionment (Clappier et al., 2017).

The computer code is still under development. In the near future we intend to include more secondary processes, such as the dependency of some reaction rates on the surface area of aerosols and a more complete description of SOA, and to develop the user interface. There is also room for significant improvements in computational efficiency, although the present version already has proven an order of magnitude faster than BF methods in some relevant situations.

*Code and data availability.* A user friendly setup for testing and reproducing the results shown in this article is available at https://doi.org/10.5281/zenodo.14162688 (Wind and Caspel, 2024). This includes a full copy of the EMEP MSC-W model code and a set of input data that can be used to produce the examples presented in this paper.

For air pollution modelling purposes, we recommend to use the official version of the full EMEP MSC-W model code and main input data available through a GitHub repository under a GNU General Public License v3.0 through https://zenodo.org/records/14507729 (EMEP MSC-W, 2024) (last access May 2025). The routines related to the Local Fractions are part of the standard model.

*Author contributions.* All authors contributed to the discussion and development of the main ideas, the applications, and the preparation of the paper. PW wrote the corresponding Fortran90 code.

*Competing interests.* The authors declare that they have no conflict of interest.

*Acknowledgements.* IT infrastructure in general was available through the Norwegian Meteorological Institute (MET Norway). Some computations were performed using resources provided by UNINETT Sigma2 – the National Infrastructure for High Performance Computing and Data Storage in Norway (grant NN2890k and NS9005k). The CPU time made available by ECMWF has been critical for the generation of meteorology used as input for the EMEP MSC-W model and the calculations presented in the current work.

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
