# Peer review of "Generalized local fractions – a method for the calculation of sensitivities to emissions from multiple sources for chemically active species, illustrated using the EMEP MSC-W model (rv5.5)"

_EGUsphere, 2024_

## Author Comment (AC2)

We thank the two anonymous referees for their careful reading of our manuscript and constructive and pertinent comments, which have undoubtedly helped to improve the quality of our work.

As a general note, we are aware that the original LF paper (Wind et al., 2020) presented the method as a way of tracking source contributions, and not as a sensitivity model. This can certainly be a cause of confusion. However, for linear species, we would argue that there is no real fundamental difference. The "generalization" presented here, instead concerns mostly the range of applicability. Compared to the first LF paper from 2020, the methodology essentially only adds the module that propagates the local fractions through the chemistry transformations. We also change units in the outputs. Where the original local fractions were dimensionless (being treated as fractions), we here multiply by the total concentration to get a concentration unit. The fundamental methodology and computer code (outside of the calculation of chemical derivatives in the chemistry module) is unchanged compared to the 2020 paper.

While not strictly necessary for the reply to reviewers, we would like to give some additional background information by elaborating on the original 2020 submission, and how this reflects on any possible confusion with regards to the LFs being a source apportionment or sensitivity methodology. In our first submission of the 2020 paper, we in fact tried to present the method as a sensitivity method, but this was not accepted by the referees (and eventually reformulated), which might be a reason for the final form to be somewhat misleading (with the original LFs being presented as a way to determine the source contributions, i.e. source apportionment, of different sources to primary particulate matter). Here is an example of one of our replies to reviewer comments from the original LF paper:

"In the future we plan to generalize the method to include chemical processes. The Local Fractions could then give information about sensitivities to changes in emissions without necessarily summing up to one hundred percent."

Referee comments for the first paper also noted the following:

"It seems to me that the local fractions deliver information about contributions, not sensitivities. Page 16, line 14: Why wouldn't the local fractions add up to 100%, and why isn't this a problem? It seems that the final sentence of the manuscript creates all sorts of problems for the interested reader. The authors could consider simply deleting this sentence."

For the current work, we also would like to stress that we aimed for this to be a technical paper. To this end, we want to document the mathematical and computational foundations of a new tool. This new tool has already been used in important applications (some of them briefly mentioned in the discussion), but each new application requires detailed discussions that we do not think fit in this methodological description paper, and which will instead be the subject of separate future publications. With this being said, we now proceed to address the comments from reviewer 1 and 2 step-by-step. Each of the comments is repeated, while our replies are highlighted in blue.

RC1: 'Comment on egusphere-2024-3571', Anonymous Referee #1, 31 Mar 2025
The authors are providing a description of how the original local fractions method has been extended making it suitable for application to pollutants influenced by non-linear processes. While the technique is quite complicated and requires some brain crunching to grasp its approach, the authors have done a good job at taking the reader by the hand through two illustrative examples, allowing easier understanding of the methodology.
The paper thereby can be considered as a good reference for a novel methodology, which provides opportunities to advance in the science of emission change sensitivities. The paper is well written and structured.
General comments:
- Since it seems the method is developed to be applied for source attribution applications
(as illustrated in the applications mentioned in the conclusions some more attention
should be paid to the applicability of the method for such applications. Explain a bit
further how this method could be used to investigate potential impacts of 15% emission
reductions, which is a common emissions reduction used for policy studies. The reader

50  is now referred to an EMEP report for the comparison with brute force at 15% emission
reduction, but it would be good to include the conclusions of that work in this paper
aswell focusing on potential differences and possible implication for the interpretation
of LF results.

55  We understood this comment to effectively come down to the question of how large the differences due to non-linearities are
between the 1 % reductions employed by the LF methodology, and the 15 % reductions typically employed by the BF method.
In part since the BF method is sometimes used to calculate (or estimate) source attribution, even though this is not the aim of
the LF method as it is presented in the current work. To provide additional context, we now refer to earlier investigations of the
impact of non-linearities by adding to Sect. 4:

60
"In particular, in EMEP Status Report 1/2024 (2024) country source-receptor matrices for yearly averages using the BF and
LF methods have been compared for several pollutants and indicators, including $O_3$ and $PM_{2.5}$. The differences are overall
less than 10 %, and can be considered as small, since they are of the same order of magnitude as methodological differences
(advection scheme and filtering) shown in Fig. 1 and Fig. 2 in the following. The differences due to non-linearities introduced
65  by reducing the emissions by 15 % in the BF method, compared to, in principle, infinitesimally small reductions in the LF
methodology, represent only a small fraction of methodological differences. We do stress that the 15 % emission reduction
employed by the BF method is in principle arbitrary (EMEP Status Report 1/2004, chapter 4, 2004), and that the differences
due to non-linearities with the LFs do not represent an actual source of methodological error."

70  In several places in the text it is mentioned that because of the small perturbations linear
assumptions can be made, what does this mean for application to investigate the impact
of potential policy measures.
Also the LF results are presented as contributions, i.e. transforming the sensitivity to a
very small emission change to a 100% contribution, please explain what is the validity
75  and limitations of this, for example are the contributions adding up to the total
concentration? This is related to the interpretation of the results for policy makers.

In effect, the interpretation of the assumptions is that the LFs give accurate results when the (policy relevant) emission
reductions are very small. For emission changes that are larger, the method as it is presented in the current work might not
80  be entirely accurate, because then non-linear effects may become important. However, for such applications, multiple LF
simulations could be performed to calculate the actual non-linear dependence on emission changes. But as alluded to in the
conclusion section, such applications will be the subject of a future publication. Nevertheless, to add further interpretation to
the LF methodology, we have added to the introduction section:

85  "The methodology is a sensitivity method. It does not directly attempt to determine the impact of any finite change of
emissions. Neither does it assign the relative total contributions from different sources (in contrast to other tagging methods
(e.g., Butler et al., 2018; Emmons et al., 2012; Dunker et al., 2002; Kwok et al., 2015; Grewe, 2013; Grewe et al., 2017; Wang et al., 1998
). Total contributions can only be directly inferred if the species are considered to have a linear dependency on emissions,
which is not the case in general. Still, due to the lower computational cost, non-linear responses can be inferred by performing
90  sensitivity analysis at several emission levels, thus providing indirectly the effect of non linear changes (EMEP Status Report 1/2024, 2024,
)."

and to Sect. 2.2:

95  "It is important to keep in mind that even if the results are presented in units of concentrations for a 100% change of
emissions, they can not be interpreted as total contributions for non-linear species. The values must be interpreted as sensitivities
to small emission changes. Those sensitivities can be both positive or negative, and will in general not sum up to total

concentrations."

100  - Throughout the paper maker sure to consistently use the word generalized in
combination with local fractions wherever appropriate to distinguish between the
original local fractions method and the newly developed generalized version.

As explained in the introduction to this author reply document, we do not consider that there is a fundamental difference:
105  The "generalization" presented here, does essentially only add the module that propagates the local fractions through the
chemistry transformations (and also a change in units in the outputs; where the original local fractions were dimensionless,
we here multiply by the total concentration to get a concentration unit). To highlight that the generalized LFs and original LFs
are fundamentally interchangeable, we have added "The generalized LF methodology (hereafter interchangeably referred to as
"LF")..." to the introduction section.

110  Specific comments:
- Introduction: explain why it is fundamental to relate the air pollution to emission sources
(i.e. for policy), add some background on how this is normally done (current practice
broader than BF) and how LF comes in.

115  To provide additional context, we have added to the introduction: "The information about the relative impact from different
countries (and possibly emission sectors) can then be used to propose an optimized set of abatement measures, as is done for
example using the GAINS (Greenhouse gas – Air pollution Interactions and Synergies) model (Amann et al., 2011)"

120  - Introduction page 1, mention also the need for tracking source sectors besides source
regions.

The sources tracked using the LFs are effectively (a collection of) sectors from any particular source region, based on how
emission input files are typically formatted. To highlight that we can track sectors from any specific source region, we have
125  added "In this work the word "source" is defined as a set of emitted species from a predefined region and from any individual
or combination of emission sector(s)."

- Introduction line 25-26, mention interaction between sources

130  We have added: "The species involved will also usually originate from different emission sources."

- Introduction line 28 – chemical species , make it clear that this concerns non-inert
chemical species

135  rewritten as "*sensitivity* (i.e., rate of change) of such active chemical species to source emission changes"

- Sections 1 and 2, as mentioned in the general comments put the method in the context
of other methods, what is the range of applicability, explain additivity limitations or
limitations with respect to the range of emission changes for which the method can be applied. What happens if there is a
140  limited chemical regime and an emission change
does not have any impact? This question relates mainly to applicability for single source
sectors or pollutants.

To add depth to the interpretation of the calculations, we have added (as discussed above): "It is important to keep in mind
145  that even if the results are presented in units of concentrations for a 100% change of emissions, they can not be interpreted
as total contributions for non-linear species. The values must be interpreted as sensitivities to small emission changes. Those

sensitivities can be both positive or negative, and will in general not sum up to total concentrations." If there is a limited chemical regime where an emission change will not have any impact, the calculated LFs will also simply be zero. In reply to other comments, we have added elsewhere in the text a note that the LFs can be either positive or negative (e.g., positive when reducing $NO_x$ emissions leads to an increase in $O_3$ due to reduced titration effects). We hope that this also further clarifies that the sensitivities may be zero in certain regimes.

- Page 3 line 66, what do you mean with second order effects, which ones? Again here you focus on small perturbations but how would you apply this method when the policy maker is interested in impact of large emission changes?

The method as is described in the current work does not directly address this (policy) question. However, the text added in reply to an above comment (repeated here), gives a more explicit explanation:

"The methodology is a sensitivity method. It does not directly attempt to determine the impact of any finite change of emissions. Neither does it assign the relative total contributions from different sources (in contrast to other tagging methods (e.g., Butler et al., 2018; Emmons et al., 2012; Dunker et al., 2002; Kwok et al., 2015; Grewe, 2013; Grewe et al., 2017; Wang et al., 1998)). Total contributions can only be directly inferred if the species are considered to have a linear dependency on emissions, which is not the case in general. Still, due to the lower computational cost, non-linear responses can be inferred by performing sensitivity analysis at several emission levels, thus providing indirectly the effect of non linear changes (EMEP Status Report 1/2024, 2024)."

- Page 3 – lines 70-73, What is the difference with the linear adjoint model? What is new to this generalized LF, why not use the existing linear adjoint model?

To highlight the differences and to provide additional context, we have added: "To our knowledge, the usage of Tangent Linear Models (TLM) has primarily been to compute backward trajectories (emission inversions (Zheng et al., 2024)), while the adjoint of TLMs is also applied in the context of data assimilation (Shankar Rao, 2007). We will here show how a TLM can be used in a 'forward mode' to compute sensitivities to emission sources in the form of, for example, source-receptor matrices."

- Page 3 – line 76, can it also be a source sector?

Indeed it can, and as described above, we have added "(from a certain sector), or in principle any other sub-class of a pollutant)" to the introduction text.

- Page 5 equation 8, explain a bit further, especially the minus sign

For clarity, in the revised manuscript the equations have been separated into two, one for the minus sign (new eq. 8) and one for calculation of deposition sensitivities (new eq. 9). An extra line of interpretation is also added.

- Page 5 line 145, 'however the range of validity is now limited', please explain what that means in practice.

As highlighted in the additions to the text described above, we now emphasize more that the method relates to the impact of very small emission changes, because then the calculated responses are effectively linear. For larger emission changes, the response will in general be non-linear, therefore being beyond the range of validity of the linear assumption. To further highlight this, we have added: "and the calculated deposition sensitivities may no longer be representative."

- Page 6 line 147, 'this may not be the case anymore', what does that mean for the

applicability of the method?

As for the previous comment, this statement reflects that for larger perturbations the response may no longer be linear, thus making the method unsuited for calculating the impact of large emission changes for (highly) non-linear species. We hope to have addressed this comment also with the reply given to the previous comment.

- Page 6 line 154, general equations: what do you mean with general? do you mean the equations for the entire method, not only chemistry? Do you show the SIA chemical transformations as example for other chemistry?

We have added that 'general equations' refers to the sensitivity equations for all species and relevant model processes. In addition, a more descriptive explanation of the general equation has been added to Sect. 2.4.2: "Eq. 25 is a general equation that describes how to update the sensitivities for any transformation. For example in the (linear) deposition case, the Jacobian is simply equal to $1 - v_i$. In the advection case (below) the indices of the concentrations would refer to pollutants at different position in space and the Jacobian will be equal to the fluxes."

- Page 6 line 175, 'those are' I guess you are referring to aij with the word those? Or to the environment, ......? So do you mean that for example the change in NO3 because of a change in SO4 is assumed constant?

Indeed this is what is meant. For added clarity, we have reformulated to "$a_i^j$ are considered constants during the time step, being independent of the scenario $k$."

- Page 7 lines 197-199, this is a difficult part, for me hard to follow, explain the small c versus capital C

The lower case is now defined in Eq. 1 as "$C_{i,k}$ represents the concentration of pollutant $C_i$ from source $k$ and we use the lower case $c_{i,k}$ to define the "dimensionless concentration", or the proportion of the pollutant that originate from source $k$.", analogous to the dimensionless fractional contributions of the original LFs.

- Page 7 lines 190-199, it made it easier for me to follow when I filled in example species, maybe an extra sentence can be added giving the example of delta SO4 as a result of delta NO3.....

While we agree that such an addition would be didactic, we also feel that it might distract from the overall flow of the text. In our view it works in the favor of the text to keep the methodological description overall analytical.

- Page 8 line 227-228 I do not understand this sentence

Reformulated as "Note that $\epsilon$ in Eq. 4 is assumed small, however $\Delta t$ or the changes in Eq. 24 are not assumed to be small. In the general case, and in the model code, the function $f$ is not assumed to be linear, and Eq. 25 is valid also for non-infinitesimal $\Delta t$. For example, in the chemical solver $\Delta t$ is divided into micro-iterations, each time step capturing the full non-linear chemistry.". We hope that this better supports the final sentence of the above quote, which was also meant to serve as a further explanation of the original (unclear) sentence referred to by the referee.

- Page 9 line 233 – what are those sets? You mentioned before 3 sets for SIA, but what would be the others, can you give some additional examples?

To clarify that sets here meant to refer to species, this line has been rewritten as "In our present implementation, 60 species will have a direct or indirect effect on other chemical species in the chemistry module. This implies that the function $f$ must be evaluated for a perturbation of each of those 60 species. In addition perturbations for emissions must be performed, but usually only a few sources will contribute for a given grid cell (typically one country, four species, and only for the lowest seven vertical levels)."

- Page 9 line 244 how are the fluxes dependent on the five nearest cells?

The exact dependence of the fluxes on concentrations in the five nearest grid cells is described in detail in the cited work of Bott (1989a, b). However, a more precise description of this dependence is difficult to explain without going more into the details of the scheme. We think such a description would distract from the current work, as the exact nature of the scheme is not necessary for the discussion in Sect. 2.5. In the context of the LFs and the equations described in this section, all that really matters is that *a* relationship exists, no matter the exact details of this relationship.

- Lines 246-247: these are nearly a duplicate of lines 244-245

Yes, removed!

- Lines 248-250, isn't a change in the advection rate also physical?

To better distinguish what we consider physical versus non-physical advection effects, we have added "These fluxes are the fraction of air masses transported to a neighboring grid cell and are in reality independent of the pollutant concentrations."

- Line 276 'improve' you have already put in italics, but isn't it more stable results?

Yes, perhaps, but since we remove a contribution that we know is wrong, we believe we can say that the result is improved.

- Line 281-283 how small/large do you expect this impact to be and in which situations can it be expected?

It is difficult to quantify as it must be disentangled from other effects, and the effect will vary largely with time, space and chemical regime. However the sum of all the differences is shown to be small in the result section, which we think suffices for making the point that these secondary effects are small.

- Line 292-293 again is expected impact small/large and in which situations can it be larger?

See answer to the above referee comment.

- Line 318-319 this is not the case for all tagging models, also in those methods they make use 'smart' computationally more efficient methods, such as calculating the loss and production rates for the total tracer concentration and then multiplying a followed by a (sparse) matrix multiplication to get the chemical conversions for each tag.

Thanks for this clarification, removed!

- Table 1 caption is fairly complicated to understand change title to relative run times generalized LF and speed up in comparison to BF. And move the sentence 'the reference time....'to the end by the time required using the LF method (...) where the reference

time is taken .....

Clarified by rephrasing to "Relative run times of LF simulations and their speed up in comparison to BF. The time unit is
defined as the time to run a single BF scenario, amounting to 29.1 seconds ($T_{\text{scenario}}$) for the model setup described in Sect. 4
(48 hours simulation on a $400 \times 260 \times 20$ grid). 50·2, 50·4, and 50·5 stands for 50 countries with 2, 4 or 5 different sector
contributions (each emitting five pollutants). The results in the table show the time to execute the LF code on eight compute
nodes. The speed up is defined as the time it would take to compute all the scenarios if a direct method (BF) had been used,
divided by the time required using the LF method ($\frac{\text{number of scenarios} \cdot T_{\text{scenario}}}{\text{total LF time}}$)."

- Table 1 total time LF total time and add unit here (reference time) to clarify

Done.

- Lines 341-342 is this for generalized LF? Which spatial resolution

To clarify, we have added and rephrased to "In practice (EMEP Status Report 1/2024, 2024), a full SR matrix calculation for
55 countries on a latitude-longitude-altitude grid with dimensions $400 \times 260 \times 20$ ($0.2° \times 0.3°$ horizontal resolution), will for
a full year LF simulation on 16 compute nodes (running 128 MPI processes on each compute node) take 18 wall time hours."

- Lines 343-344 I think the detailed numbers could be omitted, to me it would be enough
to state that in practice the LF run is 10 times faster than corresponding BF simulation,
which then agrees with table 1.

Done.

- Section 3.4 add a short introduction what this is about, are you looking for options to
simplify and obtain a speedup? Maybe change the title to optimization through
approximating chemically active species. Implemented and possible simplifications
seem to be mixed in this section, make this more clear. Are you trying to say that the
method can be sped up if you would look at S, N, .. atoms instead of species such as
NO2, NH3, .....?

To better clarify our intended purpose, we have added: "Each term of the Jacobian matrix, $\frac{\partial f_i}{\partial C_\gamma}$ in Eq. 25, is explicitly
evaluated in our present implementation. This might not be necessary, and in this section we will briefly indicate the types of
simplifications that might be implemented in the future."

- Lines 383-388 which emissions are used?

Added information that the runs were "employing standard EMEP reported emissions", but also that "The results can be
reproduced using the code and data provided under the "Code and data availability" section."

- Lines 394-397 the paper would benefit from including the conclusions of the 15%
emission reduction investigations since these are relevant to understand the
applicability (range) of the method

As noted in an earlier reply, to clarify this we have added "In particular, in EMEP Status Report 1/2024 (2024) country
source-receptor matrices for yearly averages using the BF and LF methods have been compared for several pollutants and
indicators, including $O_3$ and $PM_{2.5}$. The differences are overall less than 10 %, and can be considered as small, since

340 they are of the same order of magnitude as methodological differences (advection scheme and filtering) shown in Fig. 1 and Fig. 2 in the following. The differences due to non-linearities introduced by reducing the emissions by 15 % in the BF method, compared to, in principle, infinitesimally small reductions in the LF methodology, represent only a small fraction of methodological differences. We do stress that the 15 % emission reduction employed by the BF method is in principle arbitrary (EMEP Status Report 1/2004, chapter 4, 2004), and that the differences due to non-linearities with the LFs do not represent an
345 actual source of methodological error."

- Line 430-432 and 437-438 do you mainly see them over sea? Did you check other days, while July seems an appropriate period for ozone chemistry, maybe for SIA another time period should be preferred. And are the differences so small they would not impact in a
350 policy application?

The instabilities of the thermodynamics equilibrium module are a property of this module and not of the LF method. In the ISORROPIA-Lite equilibrium module (Kakavas et al., 2022) (also sometimes employed by the EMEP model), these instabilities can be an even more pronounced (see for example also Capps et al., 2012a; Miller et al., 2024). This is a problem
355 also for the BF method, and its effects can be large especially when compared against the impact of small emission perturbations. The LF method gives an opportunity to address this problem through the use of a filter. To clarify this, we have changed the figures to demonstrate that it is the BF method that can give poor results, and that the LF filter can detect those. We do not have the ground truth though, so we cannot show the absolute "error" of the different methods, except for that we know the results to be occasionally poor because of numerical noise and not because of physical processes.
360
- Line 438, what do you mean by these are also present in BF simulations

See answer above.

365 - Lines 468-477 here you make the connection to practical source attribution applications, I believe it is important already earlier in the document to identify the needs for this, how the assumptions and simplifications made will impact this and how to deal with larger emission reductions.

370 As noted in an earlier comment, in the introduction section we have now added "The methodology is a sensitivity method. It does not directly attempt to determine the impact of any finite change of emissions. Neither does it assign the relative total contributions from different sources (in contrast to other tagging methods (e.g., Butler et al., 2018; Emmons et al., 2012; Dunker et al., 200? )). Total contributions can only be directly inferred if the species are considered to have a linear dependency on emissions, which is not the case in general. Still, due to the lower computational cost, non-linear responses can be inferred by performing
375 sensitivity analysis at several emission levels, thus providing indirectly the effect of non linear changes (EMEP Status Report 1/2024, 2024, )."

Technical corrections
- Page 3 line 83 and 85, page 5 line 118 (also in other places in the report), please add the
380 units of e.g. Ci, E, vi.... Done.
- Line 248 add an s after emission –> emissions Done.
- Line 262 same fluxes –> same base fluxes Done.
- Line 280 'a change of emissions will change' –> may change Done.
- Line 287-288 move the word however : the impact of this limitation is however..... Done.
385 - Line 297 a test may –> a test is Done.
- Line 299 procedure may –> procedure is; iterations used may –> iterations used depends Done.
- Lines 301, 312, 317, 327, 379, LF approach –> generalized LF approach

As explained in the first part of this reply, we do not consider that there are two fundamentally different LF approaches: the "generalization" rather reflects the range of applicability.

- Line 356 this represent –> this would represent Done
- Line 363 emission –> emissions (Line 248?) Done
RC2: 'Comment on egusphere-2024-3571', Anonymous Referee #2, 07 Apr 2025
Summary and Overall Evaluation
This manuscript presents an important and methodologically sound extension of the Local Fractions (LF) approach to account for chemical transformations in chemically active pollutants, enabling efficient estimation of source sensitivities in a single simulation. The generalized LF method is implemented in the EMEP MSC-W model and validated against the brute force (BF) method. This is a highly relevant contribution to the field of atmospheric chemistry and chemical transport modelling, particularly in the context of source-receptor relationships and emission control policy support.

The paper is well-written, technically solid, and clearly illustrates the method, its implementation, and validation. The generalization of LF to chemically active species represents a significant advancement with substantial computational benefits. I recommend that the paper be accepted with minor revisions.

Major Comments

The core methodology is presented in detail and with rigor. However, the paper could benefit from including a flowchart or schematic summarizing how the generalized LF sensitivities are calculated in comparison to the BF sensitivities.

To give a clearer picture of how the implementation of the LF calculations functionally works, we have added "The LF modules are additions to the original EMEP model code, but do not affect the original results (e.g., those used for BF calculations). For each module (advection, chemistry, emissions, depositions etc.) a corresponding LF module exists, which at each time step computes the updates to the local fractions, but there is no feedback from the LF modules into the regular concentrations." to the introduction of Sect. 3.

The validation section is appropriate but could be strengthened. The comparison is limited to a 24-hour period and a single country (Germany). While this suffices for proof-of-concept, readers might benefit from including or summarizing more from the referenced EMEP Status Reports (e.g., provide direct comparisons of longer-term simulations or bias metrics).

Also in reply to comments from the first referee, we have added additional information referring to results discussed in the EMEP status reports "In particular, in EMEP Status Report 1/2024 (2024) country source-receptor matrices for yearly averages using the BF and LF methods have been compared for several pollutants and indicators, including $O_3$ and $PM_{2.5}$. The differences are overall less than 10 %, and can be considered as small, since they are of the same order of magnitude as methodological differences (advection scheme and filtering) shown in Fig. 1 and Fig. 2 in the following. The differences due to non-linearities introduced by reducing the emissions by 15 % in the BF method, compared to, in principle, infinitesimally small reductions in the LF methodology, represent only a small fraction of methodological differences. We do stress that the 15 % emission reduction employed by the BF method is in principle arbitrary (EMEP Status Report 1/2004, chapter 4, 2004), and that the differences due to non-linearities with the LFs do not represent an actual source of methodological error."

The handling of discrepancies arising from chemical regime discontinuities is important. The authors mention filtering but do not show filtered vs. unfiltered results. Including one such illustration would be helpful.

It is true that the handling of discontinuities is a major practical problem. In fact, since the submission of the original manuscript we already have developed another filtering method to give even more satisfactory results. In the revised manuscript, we take the opportunity to discuss this implementation in more details, while also including the effect of switching off the filter.

The citation of other relevant literature in the paper is done in a very narrow way. The authors tend to cite large amounts of their own work and work of their colleagues, and what seems like a bare minimum of other work from the broader literature. As a result of this, the paper lacks context about how the work fits in with other related studies. The narrow citation strategy

will also limit the discoverability of this new work through tools like citation databases. The authors should make an effort to discuss their work in relation to previous literature on adjoint / tangent linear modelling, other source attribution techniques such as tagging, and policy-relevant source/receptor modelling.

To add context, also in reply to a comment from the first referee, we have added "The methodology is a sensitivity method. It does not directly attempt to determine the impact of any finite change of emissions. Neither does it assign the relative total contributions from different sources (in contrast to other tagging methods (e.g., Butler et al., 2018; Emmons et al., 2012; Dunker et al., 2002 ). Total contributions can only be directly inferred if the species are considered to have a linear dependency on emissions, which is not the case in general. Still, due to the lower computational cost, non-linear responses can be inferred by performing sensitivity analysis at several emission levels, thus providing indirectly the effect of non linear changes (EMEP Status Report 1/2024, 2024, ." to the introduction section.

In addition, we have added a paragraph stating "To our knowledge, the usage of Tangent Linear Models (TLM) has primarily been to compute backward trajectories for emission inversions (e.g., Zheng et al., 2024), while the adjoint of TLMs is also applied in the context of data assimilation (Shankar Rao, 2007). We will here show how a TLM can be used in a 'forward mode' to compute sensitivities to emission sources in the form of, for example, source-receptor matrices." to Sect. 2.

Minor Comments and Suggestions
Line 59: how large is "large enough"? added: "larger than about ten (Sect. 3.3.2)"

Equation 1: explain why there is no subscript i on the LHS. Shouldn't the local fractions be defined per chemical species?
Line 85: shouldn't the emissions Ek also have subscript i?

We could have added a subscript $i$, however here it is assumed as part of source $k$. In the 2020 paper the source k is tracked, and therefore there is no real distinction between the source species and the pollutant species (e.g., total $PM_{2.5}$ was tracked rather than a specific chemical subspecies contained within $PM_{2.5}$). In the generalized LF case the tracked source-specific species can be different. For clarification, we have now added an example of what the indices mean in the original formulation: "For instance $i$ could refer to primary particulate matter, and $k$ could refer to primary particulate matter from the transport sector in Paris."

Equation 3: the absence of the subscript i here implies that the reduction factor applies equally to all species emitted by sector k. Is this correct, and if so, is this a limitation of the method?

$k$ refers also to an emitted species/pollutant (e.g., $NO_X$ from traffic, or benzene from traffic, etc.), and is now more generally defined as "The source $k$ can refer to any specific class of pollutants (from a certain sector, or in principle any other sub-class of a pollutant) either from a given grid cell within the tracked region (termed the 'local region') or from a predefined fixed region, such as a country."

For different chemical species emitted by a single source (in the sense of the source referring to sector such as road traffic), the calculation of the source sensitivities indeed applies the same reduction factor (perturbation) to all emitted species. But since these perturbations are taken to be infinitesimally small, this reduction factor is equally valid for all emitted species, and is not a limitation of the method.

Equation 4: Define "S" here.

added "is then defined as the sensitivity $S$, as".

Section 2.3.1: Are there any examples of this non-linear deposition in a real CTM simulation?

Yes, or at least it has been show that it is relevant. We have added reference to the paper of Fowler et al. (2001), describing a relationship between the deposition rate of $SO_2$ and $NH_3$ concentrations.

Section 2.4.1: Does the choice of a small delta limit the size of the perturbations (difference from the base case) that can be calculated using Local Fractions?

We do only attempt to evaluate the first order derivative. We have added "Mathematically a very small value can be chosen, however for numerical stability and robustness reasons in the SIA calculations, a somewhat larger value than a 1 % perturbation is preferred here." The choice of using a small perturbation does in general limit the applicability of the calculated LFs to a regime where the response to emission perturbations is approximately linear.

Line 233: how are these 60 perturbations chosen?

We have added a more explicit explanation (also in reply to a comment from the first referee) that "In our present implementation, 60 species will have a direct or indirect effect on other chemical species in the chemistry module. This implies that the function $f$ must be evaluated for a perturbation of each of those 60 species. In addition perturbations for emissions must be performed, but usually only a few sources will contribute for a given grid cell (typically one country, four species, and only for the lowest seven vertical levels). The evaluation of the function $f$ for each of these perturbations represents the most time-consuming part of the entire code"

Line 396: This strikes me as a cop-out. The whole point of the Local Fractions method is to approximate the sensitivities of non-linear systems, so some discussion of the differences between the results obtained from Local Fractions and the results of 15% perturbations is definitely appropriate in this paper.

To elaborate on this (also in reply to comments from the first referee), we have expanded the text to include "
[revised manuscript text omitted]